# High thermal conductivity in wafer-scale cubic silicon carbide crystals

Zhe Cheng [1], Jianbo Liang [2], Keisuke Kawamura[3], Hao Zhou[4], Hidetoshi Asamura[5], Hiroki Uratani[3], Janak Tiwari[4], Samuel Graham[6], Yutaka Ohno [7], Yasuyoshi Nagai[7], Tianli Feng[4], Naoteru Shigekawa[2] & David G. Cahill [1]

High thermal conductivity electronic materials are critical components for high-performance electronic and photonic devices as both active functional materials and thermal management materials. We report an isotropic high thermal conductivity exceeding 500 W m$^{-1}$K$^{-1}$ at room temperature in high-quality wafer-scale cubic silicon carbide (3C-SiC) crystals, which is the second highest among large crystals (only surpassed by diamond). Furthermore, the corresponding 3C-SiC thin films are found to have record-high in-plane and cross-plane thermal conductivity, even higher than diamond thin films with equivalent thicknesses. Our results resolve a long-standing puzzle that the literature values of thermal conductivity for 3C-SiC are lower than the structurally more complex 6H-SiC. We show that the observed high thermal conductivity in this work arises from the high purity and high crystal quality of 3C-SiC crystals which avoids the exceptionally strong defect-phonon scatterings. Moreover, 3C-SiC is a SiC polytype which can be epitaxially grown on Si. We show that the measured 3C-SiC-Si thermal boundary conductance is among the highest for semiconductor interfaces. These findings provide insights for fundamental phonon transport mechanisms, and suggest that 3C-SiC is an excellent wide-bandgap semiconductor for applications of next-generation power electronics as both active components and substrates.

Silicon carbide (SiC) plays a fundamental role in many emerging technologies such as power electronics, optoelectronics, and quantum computing[1–4]. SiC based power devices can lead a revolution in power electronics to replace Si-based technology due to its fast switching speeds, low losses, and high blocking voltage[5]. In power electronics and optoelectronics, the high localized heat flux leads to overheating of devices[6,7]. The increased device temperature degrades their performance and reliability, making thermal management a grand

challenge[6,8]. High thermal conductivity (κ) is critical in thermal management design of these electronics and optoelectronics, especially for high-power devices[9,10].

Current high κ electronic materials such as hexagonal SiC and AlN have room-temperature c-axis κ of ~320 W m$^{-1}$K$^{-1}$ for 6H-SiC, ~350 W m$^{-1}$K$^{-1}$ for 4H-SiC, and 320 W m$^{-1}$K$^{-1}$ for AlN, which are lower than metals such as silver and copper (~430 and 400 W m$^{-1}$K$^{-1}$)[11,12]. The widely used high κ value (490 W m$^{-1}$K$^{-1}$) for 6H-SiC is from Slack's

[1]Department of Materials Science and Engineering and Materials Research Laboratory, University of Illinois at Urbana-Champaign, Urbana, IL 61801, USA. [2]Department of Physics and Electronics, Osaka Metropolitan University, Sugimoto 3-3-138, Sumiyoshi, Osaka 558-8585, Japan. [3]SIC Division, Air Water Inc., 2290-1 Takibe, Toyoshina Azumino, Nagano 399-8204, Japan. [4]Department of Mechanical Engineering, University of Utah, Salt Lake City, UT 84112, USA. [5]Specialty Materials Dept., Electronics Unit, Air Water Inc. 4007-3 Yamato, Azusagawa, Nagano 390-1701, Japan. [6]George W. Woodruff School of Mechanical Engineering, Georgia Institute of Technology, Atlanta, GA 30332, USA. [7]Institute for Materials Research, Tohoku University, 2145-2 Narita, Oarai, Ibaraki 311-1313, Japan. ✉e-mail: zcheng18@illinois.edu; liang@omu.ac.jp; d-cahill@illinois.edu

measurements back to 1964 with a thermocouple-based steady-state technique[13,14]. Recent more advanced measurements based on time-domain thermoreflectance (TDTR) reduced the errors and corrected this value to ~320 W m$^{-1}$K$^{-1}$ for 6H-SiC[11,15,16], which is consistent with first-principles calculations of perfect single crystal 6H-SiC based on density functional theory (DFT)[17]. The excellent agreement of the measured 6H-SiC thermal conductivity with the predicted intrinsic thermal conductivity shows the high quality of current commercially available 6H-SiC.

Compared with the extensively studied and widely used hexagonal phase SiC polytypes (6H and 4H), the cubic phase SiC (3C) is much less well understood even though it potentially has the best electronic properties and much higher κ[1,5]. The metal oxide semiconductor field effect transistor (MOSFET) based on 3C-SiC has the highest channel mobility ever presented on any SiC polytype, which produces a large reduction in the power consumption of power switching devices[5]. 3C-SiC is a SiC polytype which can be grown on Si[5]. A long-standing puzzle about the measured κ of 3C-SiC is that the literature value is lower than that of the structurally more complex 6H phase and much lower than the theoretically predicted intrinsic thermal conductivity of 3C-SiC[1]. This contradicts the prediction of simple theory that the structural complexity and κ are inversely correlated[17]. To explain the abnormally low κ of 3C-SiC in the literature, A. Katre, *et al.* studied all the measured thermal conductivity and impurity of 3C-SiC in the literature and attributed the low κ to exceptionally strong boron defect-phonon scattering, which is even stronger than phonon scattering by vacancies[1]. 0.1% boron creates a factor of 2 decrease in κ while the same reduction is created by 2% substitutional nitrogen[1]. However, experimental validation is still lacking partly due to the challenges in growing high-quality 3C-SiC crystals[5,18]. The mature growth techniques and successful quality control of 6H-SiC crystals laid the foundation for current wide adoption of 6H-SiC electronics while the applications of 3C-SiC electronics are limited by the crystal quality and purity[5].

The potential high κ of 3C-SiC not only facilitates applications which use 3C-SiC as active electronic materials, but also enables 3C-SiC to be a thermal management material which cools devices made of other semiconductors. For thermal management materials, diamond has the highest isotropic κ among all bulk materials but is limited by its high cost, small wafer size, and difficulty in heterogeneous integration with other semiconductors with high thermal boundary conductance (TBC)[10,19,20]. Graphite has extremely strong intrinsic anisotropy in κ due to weak cross-plane van der Waals bonding[21]. The κ of carbon-based nanomaterials such as graphene and carbon nanotubes decrease significantly when assembling together or with other materials[7]. Recently, great progress has been achieved in the discovery of isotropic high κ in high-purity boron-based crystals, such as cubic BAs[22–24], natural and isotope-enriched cubic BN[25], and natural and isotope-enriched cubic BP[25–27], but all the crystal sizes are millimeter-scale or smaller. The technical difficulties in growth of high-purity large crystals prevent these high κ thermal management materials from scalable manufacturing that is required for the processing of devices. Further heterogeneous integration of these high κ thermal management materials with other semiconductors with high TBC is also challenging[28,29].

Here, we report an isotropic high κ exceeding 500 W m$^{-1}$K$^{-1}$ at room temperature in a high-purity wafer-scale free-standing 3C-SiC bulk crystal grown by low-temperature chemical vapor deposition. The measured κ agrees well with the first-principles predicted intrinsic κ of perfect single-crystal 3C-SiC. Moreover, 3C-SiC can be heterogeneously integrated with Si and AlN by epitaxial growth. The in-plane and cross-plane κ of corresponding 3C-SiC thin films are measured by beam-offset time-domain thermoreflectance (BO-TDTR). Further structural analysis such as Raman spectroscopy, X-ray diffraction (XRD), high-resolution scanning transmission electron microscopy (HR-STEM), electron back-scatter diffraction (EBSD), and second ion mass spectroscopy (SIMS)

are performed to understand the relationship between microstructure, composition, and thermal conductivity. Additionally, the TBC of 3C-SiC epitaxial interfaces with Si and AlN are studied by TDTR.

## Results

3C-SiC has a less complex crystal structure than 6H-SiC (Fig. 1a). Therefore, higher κ than 6H phase is predicted for 3C-SiC single crystal[1]. We obtain a free-standing 3C-SiC wafer (Fig. 1b) by growing 3C-SiC on a silicon substrate and then etching away the Si substrate. More details about samples can be found in Methods section. The wafer has a yellow color because of two reasons. First, the bandgap of 3C-SiC is 2.3 eV which corresponds to the energy of photons with wavelength of 539 nm. The intrinsic absorption of 3C-SiC makes it look yellow. Second, the nitrogen defects in the 3C-SiC crystal also possibly contribute to the yellow color. Peaks (795 cm$^{-1}$ for TO and 969 cm$^{-1}$ for LO) in Raman spectrum measured on the 3C-SiC crystal (Fig. 1c) agree well with the Raman peaks of 3C-SiC in the literature (796 cm$^{-1}$ for TO and 970 cm$^{-1}$ for LO)[30]. Fig. 1d shows rocking curve of the X-ray diffraction of the 3C-SiC crystal. The full width at half maximum (FWHM) of the (111) peak is 158 arcsec, showing the high crystal quality of the 3C-SiC crystal. To further probe the crystal structure of the 3C-SiC, we obtained an annular dark field STEM image (Fig. 1e) with atomically resolved lattices. The Fast Fourier transform (FFT) of the STEM image is shown in the inset of Fig. 1e. Figure 1f shows the selected area electron diffraction (SAED) pattern in a STEM, further confirming the SiC crystal is the cubic phase. More details about Raman measurements, STEM, and SAED can be found in Methods section. For extended defects, stacking faults are typically dominant in 3C-SiC compared with dislocations. The density of stacking faults of the growth surface is found to be low (about 1000 cm$^{-1}$). We performed EBSD measurements on both faces of the freestanding bulk 3C-SiC to determine the crystal orientation. The EBSD data of both the face close to Si substrate and the growth face shows single (111) orientation over the entire scanned area (2.4 mm × 0.8 mm). More details can be found in the Methods section and SI. To figure out the main impurity concentrations in 3C-SiC, SIMS was used to measure the concentrations of boron, nitrogen, and oxygen impurities. The oxygen and nitrogen concentrations measured from the growth face are 6.6 × 10$^{17}$ atoms cm$^{-3}$ and 5.8 × 10$^{15}$ atoms cm$^{-3}$, respectively. The oxygen and nitrogen concentrations measured from the face adjacent to the Si substrate before etching away Si are 2.3 × 10$^{18}$ atoms cm$^{-3}$ and 1.4 × 10$^{16}$ atoms cm$^{-3}$, respectively. The concentrations of boron impurity are below the detection limit (~3 × 10$^{13}$ atoms cm$^{-3}$) for SIMS measurements on both faces. The measured low concentrations of impurities further confirm the high quality of the 3C-SiC crystals in this work and high κ is expected[1]. The other point defects such as vacancies were not characterized due to technical difficulties but we expect low concentrations of them.

We performed TDTR measurements on the free-standing 3C-SiC bulk crystal from the growth face to obtain its thermal conductivity. Figure 2a shows an example of the TDTR ratio data (circles) and model fitting (solid line) for the bulk 3C-SiC sample with 5× objective and 9.3 MHz modulation frequency. The dash lines are model curves using κ 10% larger or 10% smaller than the best-fit κ to illustrate the measurement sensitivity. More details about the TDTR measurements can be found in the Methods section and SI. To evaluate the effect of ballistic thermal transport on TDTR measurements of high κ samples, we did multiple TDTR measurements with different spot sizes (10.7 μm for 5× objective, 5.5 μm for 10× objective, and 2.7 μm for 20× objective) and different modulation frequencies (1.9–9.3 MHz). We observed weak dependence of measured κ on the modulation frequency (Fig. 2b) while strong reduction in the measured κ for 20× compared to 5× and 10× (Fig. 2b). This reduction is due to the ballistic thermal transport in the sample and the mismatch in the distributions of phonons that carry heat across the metal transducer-sample interface

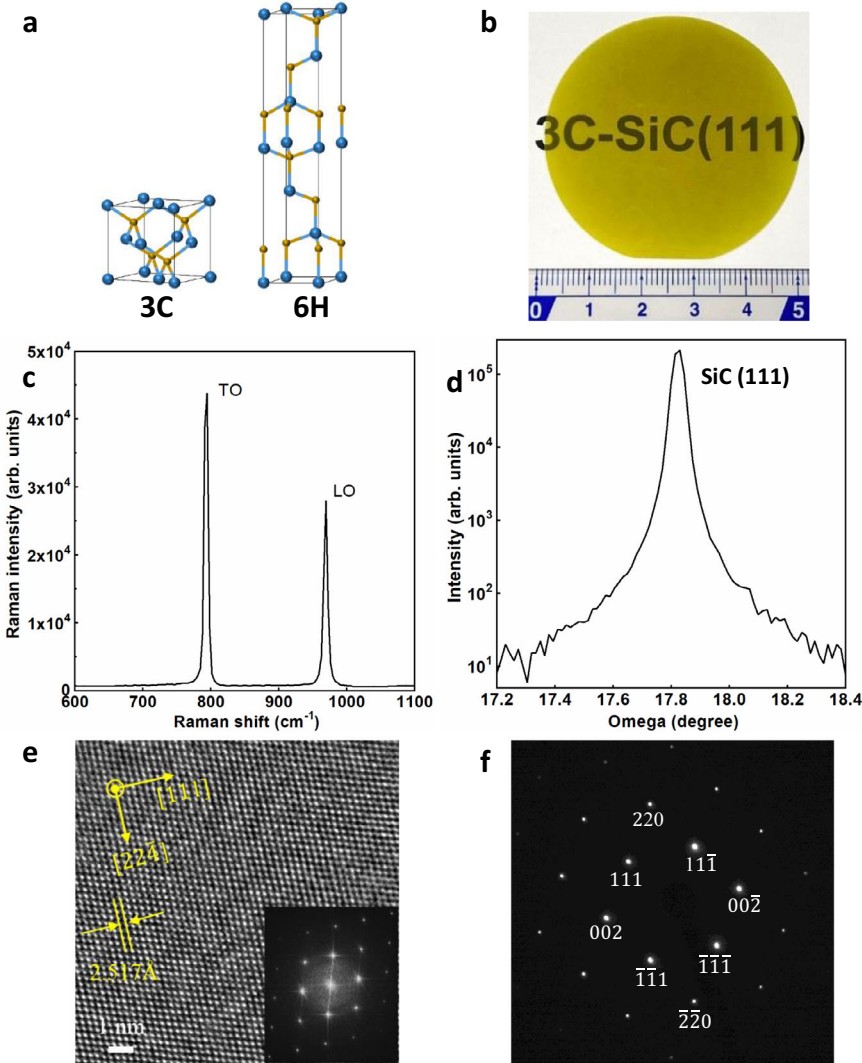

**Fig. 1 | Structure of wafer-scale free-standing 3C-SiC bulk crystals. a** Atomic structures of 3C-SiC and 6H-SiC. **b** Picture of a 3C-SiC 2-inch wafer. The unit of the ruler is cm. **c** Raman spectrum of 3C-SiC crystal. **d** X-ray diffraction (XRD) of 3C-SiC. **e** High-resolution STEM image of 3C-SiC taken along the [$\bar{1}$10] zone axis. The inset: Fast Fourier transform (FFT) of the STEM image. **f** Selected area electron diffraction pattern of 3C-SiC taken in the [$\bar{1}$10] zone axis.

and in the sample[31]. We used 9.3 MHz and 5× objective for the remainder of the measurements on the κ of bulk 3C-SiC (Figs. 2c, 3). The thickness of the free-standing 3C-SiC bulk crystal (100 μm) is much larger than the dominant phonon mean free paths in 3C-SiC and the thermal penetration depth in the TDTR measurements (the phonon dispersion relation and phonon mean free path accumulated thermal conductivity of perfect 3C-SiC single crystal calculated by DFT are included in the SI).

The measured κ of 3C-SiC at room temperature is compared with other high κ crystals as a function of wafer size (Fig. 2c)[11,12,15,16,20,22,25,26]. The recently reported boron-based crystals have high κ but the achievable crystal sizes are millimeter-scale or smaller. Single crystal diamond has a larger wafer size, up to 2 inch, but wide-range adoptions are limited by the high cost and difficulty in heterogeneous integration with other semiconductors[19,20,29]. Heterogeneous epitaxial growth of single crystal diamond on Si and GaN is challenging[29]. Current chemical vapor deposited (CVD) polycrystalline diamond results in significantly reduced and anisotropic κ[32,33].

The 3C-SiC wafer reported in this work can reach up to 6-inch in size with an isotropic high κ exceeding 500 W m$^{-1}$K$^{-1}$. The measured κ of 3C-SiC is higher than all metals and the second highest among all large crystals (only surpassed by single crystal diamond). The κ of 3C-

SiC at room temperature is ~50% higher than the c-axis κ of 6H-SiC and AlN, and ~40% higher than the c-axis κ of 4H-SiC.

We further measured the κ of bulk 3C-SiC crystal at high temperatures. The measured temperature dependent κ of bulk 3C-SiC is compared with previously measured κ values in the literature, κ values of perfect single crystal predicted by DFT, and that of other high κ crystals (See Fig. 3a, b). The measured κ agrees well with DFT-calculated κ of perfect single crystal 3C-SiC at all measured temperatures. The measured κ in this work is >50% higher than the literature values of 3C-SiC at room temperature, and surpasses that of the structurally more complex 6H-SiC. These results are consistent with the theoretical calculations that structural complexity and κ are inversely related[17]. The measured high κ resolves a long-standing puzzle about the abnormally low κ values in the literature which was attributed to the extrinsic defect-phonon scatterings in 3C-SiC[1]. Boron defects in 3C-SiC cause exceptionally strong phonon scatterings which results from the resonant phonon scattering by the boron impurity[1]. The measured boron impurity concentration is negligible (below the detection limit: $3 \times 10^{13}$ atoms cm$^{-3}$) in our 3C-SiC crystals according to the SIMS measurements. The oxygen and nitrogen concentrations are also low ($6.6 \times 10^{17}$ atoms cm$^{-3}$ and $5.8 \times 10^{15}$ atoms cm$^{-3}$). The rocking curve of XRD measurements shows a full width at half maximum

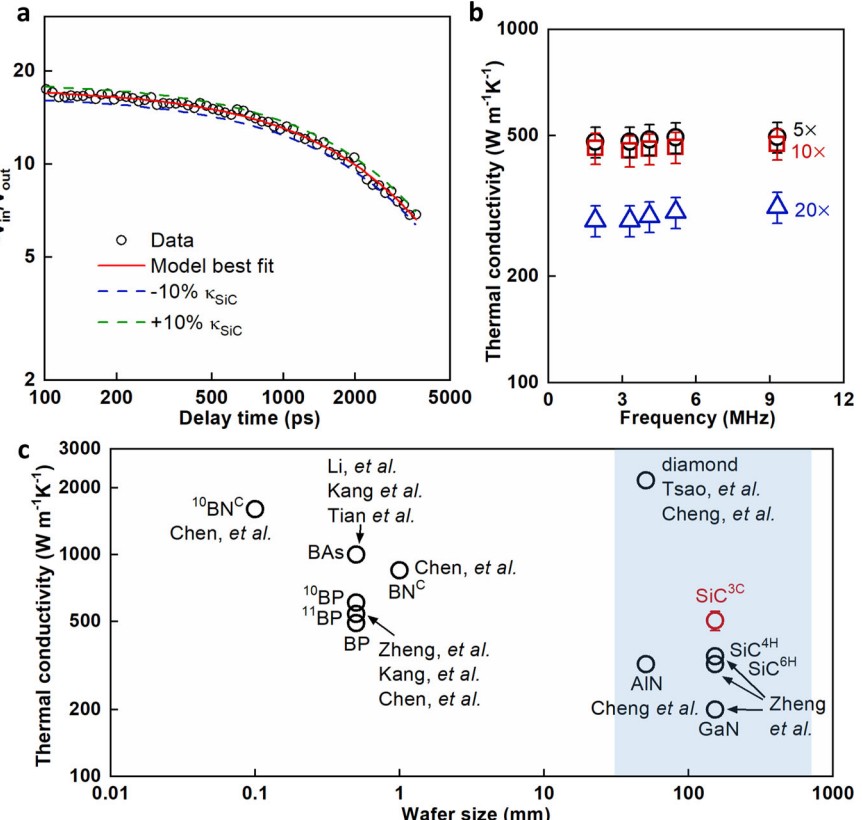

**Fig. 2 | High κ of 3C-SiC bulk crystals near room temperature. a** TDTR ratio data (circles) and model fitting (solid line) for 3C-SiC sample. The dash lines are model curves using κ 10% larger or 10% smaller than the best-fit κ to illustrate the measurement sensitivity. **b** Dependence of modulation frequency and laser spot size on the measured κ of 3C-SiC near room temperature. The definition of error bars can be found in SI. **c** The measured κ of 3C-SiC at room temperature is compared with other high κ crystals as a function of wafer size[11,12,15,16,20,22,25,26]. The shaded part includes the data of large crystals.

(FWHM) of 158 arcsec. Both crystal quality and crystal purity affect thermal conductivity. Both the high-purity and good crystal quality of our 3C-SiC crystals result in the observed high κ. The high κ in this work validates the theory proposed in the literature that the abnormally low κ observed in the literature is a consequence of the defective, polycrystalline quality of the 3C-SiC samples instead of the intrinsic property of 3C-SiC[1].

To further prove that the B impurity reduces thermal conductivity significantly as predicted by the theoretical paper, we grow an additional sample (3C-SiC film on Si substrate) which is intentionally doped with B. The concentration of the intentional boron doping is $1-2 \times 10^{19}$ atoms cm$^{-3}$ and the thickness of the 3C-SiC film is 1.87 μm. The measured thermal conductivity of this doped sample is 324 W m$^{-1}$ K$^{-1}$ which is about 20% smaller than the un-doped 3C-SiC film with a similar thickness (1.75 μm in the main text). This reduction in thermal conductivity of 3C-SiC is consistent with the theoretical prediction in ref. 1., which further supports our claim about the B defects and thermal conductivity.

We also compare the measured temperature dependent κ of bulk 3C-SiC crystals with that of AlN, 6H-SiC, and GaN. We include both the in-plane κ and cross-plane κ of 6H-SiC since the κ of 6H-SiC is anisotropic. The DFT-calculated κ values of perfect single crystals agree well with the measured κ values and both are proportional to the inverse of temperature due to the dominant phonon-phonon scatterings in these crystals at high temperatures. The measured κ values of 3C-SiC are 2.5 times as high as that of GaN, making 3C-SiC a potential candidate as substrates of GaN-based power electronics. The high κ of 3C-SiC will motivate the study of power electronics which use 3C-SiC as active device material as a more advanced addition to currently wide-adopted 4H-SiC and 6H-SiC.

We performed beam-offset time-domain thermoreflectance (BO-TDTR) on 3C-SiC thin films grown on Si substrates to obtain the in-plane κ of 3C-SiC films[34,35]. During BO-TDTR measurements, the pump beam is offset relative to the probe beam, as shown in Fig. 4a. An example of the out-of-phase TDTR signal on a 2.52-μm-thick SiC film on Si sample is shown as a function of the beam offset distance. The full width at half maximum (FWHM) is a measure of the lateral heat spreading which is used to fit for the in-plane κ of the 3C-SiC thin film. More details about the BO-TDTR can be found in the Methods section and SI. The measured in-plane thermal conductivity of 3C-SiC thin films are lower than that of the bulk 3C-SiC crystal due to the size effect. The measured in-plane κ values of 3C-SiC thin films at room temperature are compared with that of other close-to-isotropic high κ thin films such as AlN, diamond, and GaN (see Fig. 4b; strongly anisotropic materials graphite and h-BN have high in-plane κ values but we do not include them here). The in-plane κ of 3C-SiC thin films show record-high values, even higher than that of diamond thin films with equivalent thicknesses. We attribute these high in-plane κ values to the high-quality of the 3C-SiC thin films. These high in-plane κ values of 3C-SiC thin films facilitate heat spreading of localized Joule-heating in power electronics.

The cross-plane κ of the 3C-SiC thin films are measured by TDTR. The dependence of cross-plane κ on film thickness and temperature are shown in Fig. 4c, d. The measured cross-plane thermal conductivity of 3C-SiC thin films are lower than that of the bulk 3C-SiC crystal due to size effect. For the phonons in 3C-SiC with mean free paths longer than the film thickness, the phonons scatter with the film boundaries which cause reduction in the phonon mean free paths and corresponding thermal conductivity. The cross-plane κ of 3C-SiC thin films are among the highest values ever known, even higher than or comparable to that

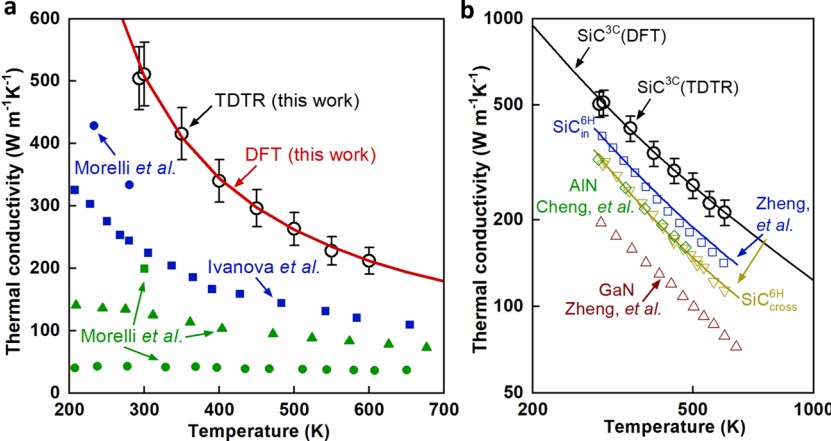

**Fig. 3 | Temperature dependent κ of bulk 3C-SiC crystals. a** Comparison of the measured κ in this work with previous measured κ in the literature[1,50,51]. The κ value (red line) predicted by density functional theory (DFT) in this work is also included[1,52]. The definition of error bars can be found in SI. **b** Comparison of temperature dependent κ of 3C-SiC with c-axis κ of bulk 6H-SiC, AlN, and GaN[1,11,12]. The symbols are experimentally measured values while the lines are DFT-calculated values of perfect single crystals[1,11]. We include both the cross-plane κ and the in-plane κ of 6H-SiC since its κ is anisotropic.

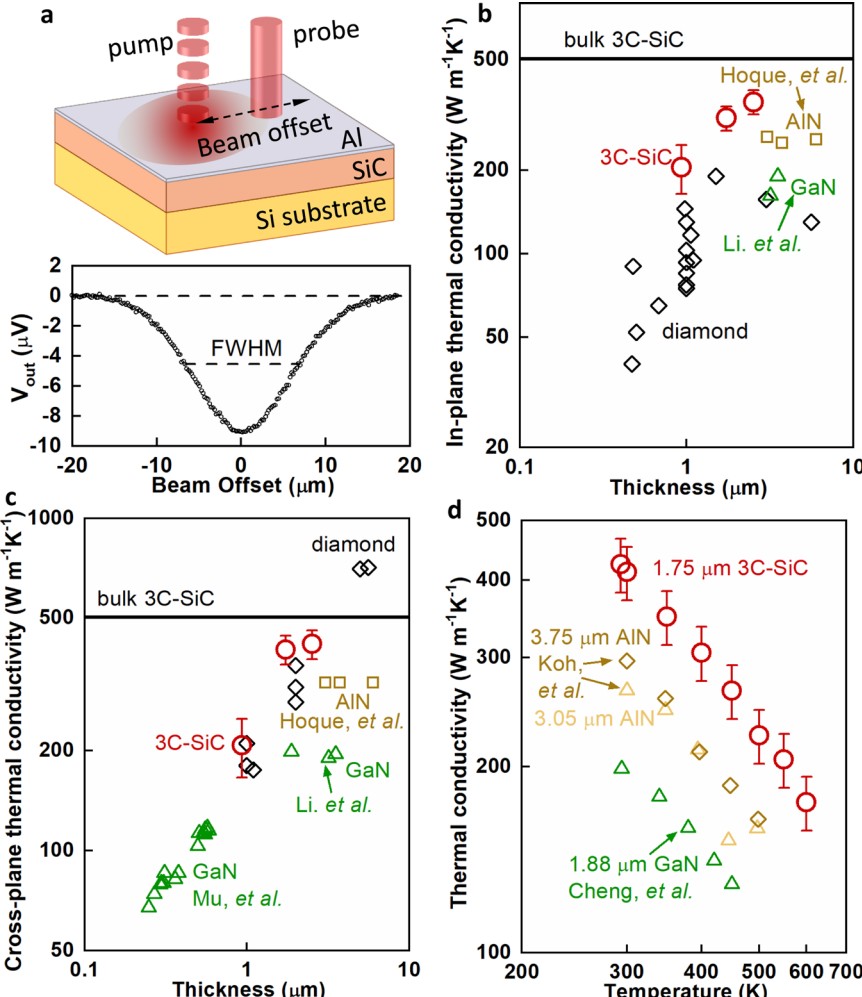

**Fig. 4 | High in-plane and cross-plane κ of 3C-SiC thin films. a** Beam-offset TDTR technique for in-plane κ measurements. The out-of-phase TDTR signal on a 2.52-μm SiC on Si sample is shown as a function of beam offset distance. **b** In-plane κ of 3C-SiC thin films. The κ of 3C-SiC bulk crystal and the in-plane κ of other close-to-isotropic high κ thin films are also included for comparison[32,53–59]. The definition of error bars can be found in SI. **c** Cross-plane κ of 3C-SiC thin films. The κ of 3C-SiC bulk crystal and cross-plane κ of other high κ thin films are also included for comparison[20,32,37,40,53,54,56,59–61]. **d** Temperature dependent cross-plane κ of a 1.75-μm-thick 3C-SiC thin film. The temperature dependent cross-plane κ of AlN and GaN thin films are also included[20,61].

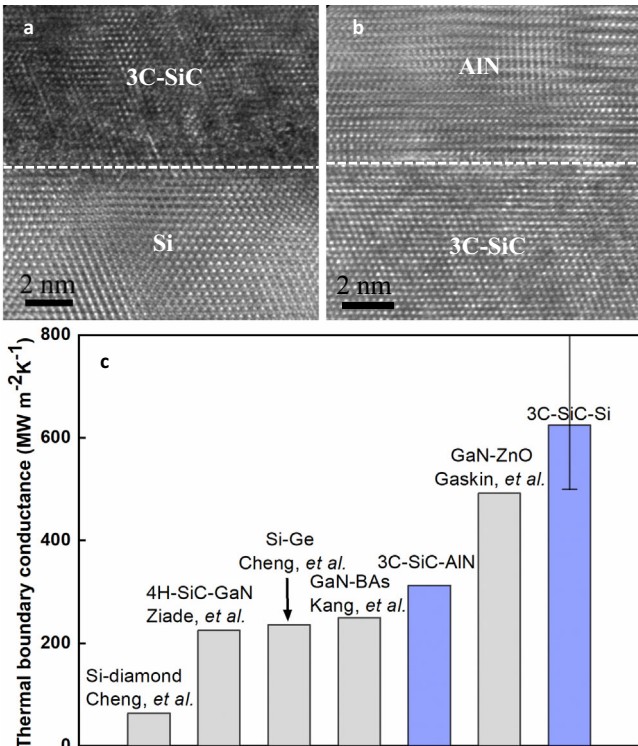

**Fig. 5 | High thermal boundary conductance of 3C-SiC epitaxial interfaces. a** TEM cross-section image of 3C-SiC-Si interfaces. **b** TEM cross-section image of 3C-SiC-AlN interfaces. **c** Thermal boundary conductance of 3C-SiC-Si interfaces and 3C-SiC-AlN interfaces. The TBC values of other semiconductor interfaces are included for comparison[28,36–38,40,41]. The definition of error bars can be found in SI.

of diamond thin films with equivalent thicknesses. The cross-plane κ of 1.75-μm-thick 3C-SiC reaches ~80% of the κ of bulk 3C-SiC, up to twice as high as the κ of bulk GaN. Even the 0.93-μm-thick 3C-SiC film has a cross-plane κ close to that of bulk GaN. The different tendency of the thickness dependent thermal conductivity for different semiconductors are due to the different intrinsic phonon mean free path distributions in these semiconductors. The calculated phonon mean free path accumulated thermal conductivity of 3C-SiC scaled by the bulk thermal conductivity is compared with other semiconductors (see SI). It is notable that, since the epitaxial diamond thin films are polycrystalline, the grain boundaries and other structural imperfections also scatter phonons and affect the tendency of thermal conductivity as a function of film thickness. Figure 4d compares the temperature dependent cross-plane κ of some wide-bandgap semiconductor thin films. In the measured temperature range, all the cross-plane κ values of 3C-SiC are higher than that of AlN and GaN with even larger thicknesses. The high cross-plane κ, combined with the high in-plane κ, of these 3C-SiC thin films make them the best candidate for thermal management applications which use thin films.

The epitaxial growth of 3C-SiC not only produces high-quality thin films which have high in-plane and cross-plane κ values, but also creates high-quality heterogeneous interfaces which are potentially thermally conductive. The cross-section TEM images of the epitaxial 3C-SiC-Si and 3C-SiC-AlN interfaces are shown in Fig. 5a, b to study the interfacial structure. Their TBC are measured by TDTR and compared with that of other semiconductor interfaces (Fig. 5c). All the interfaces are smooth interfaces with sub-nm roughness. Only the BAs-GaN interface is a bonded interface with a thin amorphous interfacial layer while all the other interfaces are fabricated by growing one semiconductor on top of the other well-polished semiconductor substrate. The measured 3C-SiC-Si TBC (~620 MW m$^{-2}$K$^{-1}$) is among the highest

values for all interfaces making up of semiconductors[36], about ten times as high as that of the diamond-Si interfaces[37], about 2.5 times as high as that of epitaxial Si-Ge interfaces[38]. It also approaches the maximum TBC of any interface involving Si, which is only limited by the rate that thermal energy in Si can impinge on the crystallographic plane[39]. The measured 3C-SiC-AlN TBC is higher than the GaN-BAs TBC and 4H-SiC-GaN TBC[28,40,41]. These high TBC values of 3C-SiC related interfaces facilitate heat dissipation of electronics and optoelectronics which use 3C-SiC, especially for the cases with an increasing number of interfaces as the minimization of devices.

In summary, this work reported an isotropic room-temperature high thermal conductivity exceeding 500 W m$^{-1}$K$^{-1}$ in high-purity wafer-scale free-standing 3C-SiC bulk crystals, which is ~50% higher than commercially available 6H-SiC and AlN. It is >50% higher than the previously measured κ of 3C-SiC in the literature, and is the second highest among large crystals. We also studied the κ of corresponding 3C-SiC thin films and found record-high in-plane and cross-plane κ values. The measured higher κ of 3C-SiC than that of the structurally more complex 6H-SiC validates that structural complexity and κ are inversely related, resolving a long-standing puzzle about the perplexingly low κ of 3C-SiC in the literature. Impurity concentrations measured by SIMS revealed the high-purity of our 3C-SiC crystals and the XRD measurements revealed the good crystal quality of our 3C-SiC crystals. Both contribute to the observed high κ. Furthermore, high TBC values were observed across epitaxial 3C-SiC-Si and 3C-SiC-AlN interfaces. The measured 3C-SiC-Si TBC is among the highest for semiconductor interfaces, about ten time as large as that of diamond-Si interfaces. The high κ observed in 3C-SiC bulk crystals and thin films, combined with the high TBC of epitaxial 3C-SiC interfaces, suggest 3C-SiC an excellent candidate for applications of next-generation power electronics and optoelectronics. 3C-SiC has the best thermal (highest thermal conductivity) and electrical (highest channel mobility) properties among all SiC polytypes, and is a polytype which can be grown on Si which enables integration of 3C-SiC electronics with Si electronics with exceptionally high thermal boundary conductance.

## Methods
### Samples
The 3C-SiC samples in this work are acquired from Air Water Inc. and are available for purchase. The 3C-SiC crystals are grown on (111) Si substrates by low-temperature chemical vapor deposition (LT-CVD) in a customized CVD reactor at 1300 K. The orientation of the Si substrate and the growth temperature are important to grow high-quality crystals. Since both Si and 3C-SiC have the same rotational symmetry (120°) about the [111] axis, (111) 3C-SiC layers can be grown on (111) Si substrates with low density of stacking faults and double positioning boundary at relatively low crystal growth temperature (1300 K). The free-standing bulk 3C-SiC crystal is obtained by growing ~100-μm-thick 3C-SiC on Si substrates and then etching away the Si substrates by HNA (HF: HNO$_3$: H$_2$O). The stacking faults density observed on the growth face is about 1000 cm$^{-1}$ according to cross-sectional TEM study. The thermal conductivity of thermally thick 3C-SiC films grown on (100) Si purchased from MTI is only 90 W m$^{-1}$ K$^{-1}$, which is significantly lower than that of our samples[42].

### Thermal characterizations
The κ and TBC are measured by time-domain thermoreflectance (TDTR). We coat ~90-nm-thick Al on the to-be-measured sample as TDTR transducer before TDTR measurements. TDTR is an ultra-fast laser based pump-probe technique which can measure thermal properties of both bulk and nanostructured materials[33,43]. A modulated pump laser beam heats the sample surface periodically while a delayed probe laser beam detects the temperature variations of the sample surface via thermoreflectance. The signal picked up by a photodetector and a lock-in amplifier is fitted with an analytical heat

transfer solution of the sample structure to infer the unknown parameters (for example, κ of 3C-SiC and TBC of the metal transducer-SiC interface when measuring the 3C-SiC bulk crystals). We used 5× objective (spot size 10.7 μm) and 9.3 MHz when measuring the κ of the 3C-SiC bulk crystals and the cross-plane κ of 3C-SiC thin films. The growth face of the bulk 3C-SiC is polished and TDTR is performed on the growth face. When measuring the 3C-SiC thin films, the thicknesses of Al transducer and 3C-SiC thin films are measured by picosecond acoustic technique[44]. More details about the thickness measurements and used literature values of heat capacity can be found in SI. The in-plane κ of 3C-SiC thin films are measured by BO-TDTR with a modulation frequency of 1.9 MHz and an objective of 10×[34,35]. We also used the 5× objective to repeat the BO-TDTR measurements and obtained consistent results.

### Raman spectroscopy
Raman measurements were performed on the 3C-SiC bulk crystal with a Horiba LabRAM confocal Raman spectroscopy imaging system. The used laser wavelength is 532 nm. The acquisition time is 600 s and the objective is 50×.

### SIMS characterizations
The depth profiles of the O, N, and B atomic densities on the face close the Si substrate and the growth face were analyzed by secondary ion mass spectrometry (SIMS) (CAMECA; IMS-4f). An area of 150 μm × 150 μm was sputtered with a beam of $O_2^+$ beam accelerated at 8 keV to obtain the depth profile of the B atomic density, an area of 220 μm × 220 μm was sputtered with a beam of $C_s^+$ beam accelerated at 14.5 keV to obtain the depth profiles of the O and N atomic densities.

### STEM and SAED measurements
Scanning transmission electron microscopy (STEM) and selected area electron diffraction (SAED) (JEM-2200FS; JEOL) were used to analyze the crystal quality of the 3C-SiC crystals and the interfaces at an acceleration voltage of 200 kV. TEM samples were prepared by using a focused ion beam (FIB) system (Helios NanoLab 600i DualBeam; Thermo Fisher Scientific) by depositing a protective layer and milling using a 30 kV accelerating voltage, and final etching using a 2 kV accelerating voltage at room temperature.

### XRD measurements
The crystal quality of the 3C-SiC crystals was characterized by the full width at half maximum on the X-ray rocking curve of the 3C-SiC (111) peak using an X-ray diffraction system (D8 Discover; Bruker). A Cu-Kα X-ray source accelerating at 40 kV with a current of 40 mA was applied to record the XRD patterns in the range of 17.2–18.4° with a step of 0.015°. An incident slit with a width of 2 mm and a collimator with a diameter of 0.1 mm were used.

### EBSD measurements
The crystal direction of the 3C-SiC crystals was analyzed by an Electron Backscatter diffraction (EBSD) system (FE-SEM JSM-6500F; JEOL) with a high-resolution scanning electron microscope (SEM) and a TSL orientation imaging microscopy (OIM) analyzer. The SEM was operated at 20 kV, and a scan area of 2.4 mm × 0.8 mm was performed using a hexagonal grid with a step size of 2 μm. The EBSD measurements are done at Toray Research Center, Inc.

### First-principle calculations
DFT simulations were performed by using the Vienna Ab initio simulation package (VASP) with the projector-augmented-wave method and the local density approximation (LDA) for exchange and correlation[45–47]. The plane-wave energy cutoff is selected as 500 eV. The primitive cell is relaxed with the energy convergence threshold of $10^{-8}$ eV, force convergence threshold of $10^{-7}$ eV/Å, and k-mesh of

15 × 15 × 15. The obtained lattice constant is 4.3306 Å. In the second-order and third-order force constants calculations, using Phonopy and ThirdOrder[48,49], the supercell size is selected as 5 × 5 × 5 (250 atoms) with a 3 × 3 × 3 k-mesh and energy convergence threshold of $10^{-8}$ eV. The non-analytical correction that splits LO and TO phonons at Γ point is considered in the phonon dispersion calculations. Up to the 6th nearest neighbor of atoms are included in the third-order force constants extraction. The temperature-dependent thermal conductivity and phonon mean free path accumulated thermal conductivity are calculated by using ShengBTE using a 36 × 36 × 36 phonon q-mesh and a broadening factor of 0.1[49]. The calculation convergence regarding q-mesh and broadening factor is studied. Natural isotope-phonon scattering is included in the calculations.

### Reporting summary
Further information on research design is available in the Nature Portfolio Reporting Summary linked to this article.

## Data availability
The datasets generated during and/or analyzed during the current study are available from the corresponding authors upon reasonable request.

## Code availability
The code used for calculations, simulations, and data analysis is available from the corresponding authors upon reasonable request.

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

## Acknowledgements

Z.C. and D.G.C. acknowledge the financial support from an endowed position created by the Grainger Engineering Breakthroughs Initiative. Z.C. and D.G.C. thank Guangxin Lyu for help of Raman measurements.

The fabrication of the TEM samples was performed at The Oarai Center and at the Laboratory of Alpha-Ray Emitters in IMR under the Inter-University Cooperative Research in IMR of Tohoku University (NO. 202112-IRKMA-0016). The observation of the TEM samples was supported by Kyoto University Nano Technology Hub in the "Nanotechnology Platform Project" sponsored by the Ministry of Education, Culture, Sports, Science and Technology (MEXT), Japan. H.Z., J.T., and T.F. acknowledge the support from National Science Foundation (NSF) (award number: CBET 2212830). The computation used resources of the National Energy Research Scientific Computing Center, a DOE Office of Science User Facility supported by the Office of Science of the U.S. Department of Energy under Contract No. DE-AC02-05CH11231 using NERSC award BES-ERCAP0022132. The support and resources from the Center for High Performance Computing (CHPC) at the University of Utah and the Advanced Cyberinfrastructure Coordination Ecosystem: Services & Support (ACCESS) of NSF are gratefully acknowledged. S.G. acknowledges the financial support from U.S. Office of Naval Research under a MURI program (Grant N00014-18-1-2429).

## Author contributions

Z.C. initialized this project, developed the idea, and finished all the thermal measurements. Z.C. and J.L. coordinated the project. J.L. polished the bulk 3C-SiC samples and performed TEM studies. K.K., H.A., and H.U. grew the 3C-SiC samples and did XRD measurements. H.Z., J.T., and T.F. did the first principle calculations. Y.O. and Y.N. prepared the TEM FIB samples. Z.C. wrote the manuscript with inputs from all authors. J.L. assisted with manuscript preparation. S.G. and N.S. commented on the manuscript. D.G.C. provided overall guidance to the project and reviewed the manuscript.

## Competing interests

K.K H.A. and H.U. are employees of Air Water, Inc. which sells 3C-SiC related products. All the other authors declare no competing interest.
