## [Peer Review File · Nature Communications]

High Thermal Conductivity in Wafer-Scale Cubic Silicon Carbide CrystalsREVIEWER COMMENTS

Reviewer #1 (Remarks to the Author):

To the Editors/Authors,

The manuscript reported an isotropic high thermal conductivity exceeding $500 \text{ Wm}^{-1}\text{k}^{-1}$ at room temperature in high-quality wafer-scale 3C-SiC, which may be more accurate measurement of thermal conductivity of pure 3C-SiC. The corresponding 3C-SiC thin films were also found to have high in-plane and cross-plane thermal conductivity. The thermal boundary conductance of 3C-SiC/Si interface was found highest in some semiconductor interfaces. These findings should help to promote the application of 3C-SiC in the wide-bandgap semiconductor. However, in my opinion the manuscript in current version is not suitable to be published in this Journal.

There are some concerns about this work, as described below.

(1) The authors stated that their results resolve the puzzle of literature experiment values of thermal conductivity of 3C-SiC and 6H-SiC: The former is lower than the latter, but the structure of the latter is more complex. The conclusion may be unquestionable, but the authors attributed this to the high purity and high crystal quality of their 3C-SiC sample. However, the purity and quality of 6H-SiC may also not be guaranteed in the literature. On the other hand, the authors attribute the low thermal conductivity in the literature to the high B content in the sample, where the inference is fallacious without reasonable evidence to support it.

(2) Generally speaking, the thermal conductivity of thin film is usually higher than that of bulk sample due to size effect. The authors should assess the effect of small size and low dimension on the main conclusion of this work.

(3) The authors attributed high thermal conductivity of 3C-SiC to the lack of defect-phonon scatterings. However, there was characterization missing for the defects in the 3C-SiC samples.

(4) The labeling of the references in Figs. 2-5 is confusing and unclear to the readers.

Reviewer #2 (Remarks to the Author):

In this manuscript, the authors report a variety of thermal conductivity measurements on high-purity bulk and thin-film 3C-SiC samples grown by low-temperature CVD. They find extremely high thermal conductivity values which are to be expected based on theory. However, as the authors point out, previous measurements on 3C-SiC materials (of presumably poorer quality) had shown lower thermal conductivity than predicted by theory. Thus, this work appears to resolve that contradiction between the theoretical predictions and previous experiments.

The work is interesting (a) from a purely scientific standpoint (resolving the aforementioned contradiction between theory and experiments), and also (b) since 3C-SiC has the potential for widespread use in high power electronic devices due to its excellent thermal and electrical properties and compatibility with other important materials.

The work is complete with thermal conductivity measurements of bulk materials, thin-films of various thickness (both in-plane and cross-plane), and also as a function of temperature. Thermal boundary conductance measurements are also reported. The corresponding structural characterization is also extensive and complete. The thorough work makes for a convincing argument that these measurements are correct.

Overall, the manuscript is well-written with extensive references to the relevant literature. It should be well received and highly cited within the community. I only have a few small comments below which should be easy to address or dismiss.

Comments

- Small point, but SAED should be defined when first used in line 109.
- It might be worthwhile to state the detection limit in SIMS for boron in line 120. This would help readers place an upper limit on the boron concentration in the samples, which is helpful in context with the calculations from Ref. 1.
- In lines 105, 184, 212, 327 “magnitude” should be “maximum”
- This may well be outside the scope of the work, or proprietary company information, but as I read the manuscript I found myself thinking that other groups have likely (surely?) tried to create high quality (low boron) 3C-SiC before. So the natural question to me is, what was different in the growth of these materials that makes them so much better? A few sentences or a short paragraph regarding the improvements in growth would probably be welcome to most readers.
- The word “properties” (or a similar substitute) appears to be missing from line 279.

Reviewer #3 (Remarks to the Author):

This manuscript reported a high thermal conductivity in cubic-SiC. The value of over 500 W/mK at room temperature is probably the highest value for SiC till now. It also confirm the traditional theory that structural complexity and thermal conductivity are inversely correlated. I think that the paper can satisfy the journal after revision. Please find questions as follows:

1. Frankly speaking, it is not certain that 3C-SiC is the only SiC polytype which can be grown on Si.
2. The 2 inch wafer of the 3C-SiC shows yellow color. What kind of defects or dopants make the wafer looks yellow? How does those defects influence the thermal conductivity?
3. The author claimed that Boron greatly influenced the thermal conductivity of 3C-SiC. In my understanding, as a point defect, Boron defects usually don't degrade the crystal quality much, in particular the values of XRD omega-rocking. In other words, a high concentration of boron does not prevent SiC as a single crystal. Therefore it is better to make it clear that the excellent value reported in this work is due to the single crystallinity or the quite low concentration of Boron?
4. Nowadays, the single crystal diamond is larger than 1 inch.
5. In page 13, 'thin films such AIN' should be 'thin films such as AIN'
6. In fig.4, the authors compared the cross-plane thermal conductivity of different kinds of semiconductors with varied thicknesses. What is the reason for the thickness-dependent thermal conductivity? Why do the 3C-SiC, AIN, GaN and diamond show different tendency?
7. In fig. 5, The author compared the thermal boundary conductance of different kinds of interface. It is known that the boundary conductance is greatly related the interface roughness? How about the interface sharpness of those interfaces?

We sincerely thank the reviewers and the editor for the precious time and attention on our manuscript. We also greatly appreciate the constructive and valuable comments from the reviewers. These insightful suggestions and corresponding revisions improve our manuscript and make it more complete.

Reviewer #1:

The manuscript reported an isotropic high thermal conductivity exceeding 500 W/m-K at room temperature in high-quality wafer-scale 3C-SiC, which may be more accurate measurement of thermal conductivity of pure 3C-SiC. The corresponding 3C-SiC thin films were also found to have high in-plane and cross-plane thermal conductivity. The thermal boundary conductance of 3C-SiC/Si interface was found highest in some semiconductor interfaces. These findings should help to promote the application of 3C-SiC in the wide-bandgap semiconductor. However, in my opinion the manuscript in current version is not suitable to be published in this journal.

Response: We thank the reviewer for the positive evaluation of our manuscript as “help promote the application of 3C-SiC in the wide-bandgap semiconductor”.

Comment (1): The authors stated that their results resolve the puzzle of literature experiment values of thermal conductivity of 3C-SiC and 6H-SiC: The former is lower than the latter, but the structure of the latter is more complex. The conclusion may be unquestionable, but the authors attributed this to the high purity and high crystal quality of their 3C-SiC sample. However, the purity and quality of 6H-SiC may also not be guaranteed in the literature. On the other hand, the authors attribute the low thermal conductivity in the literature to the high B content in the sample, where the inference is fallacious without reasonable evidence to support it.

Response: We thank the reviewer for the comment which motivates us to further clarify the mentioned points and avoid confusing readers. Please note that we include the intrinsic thermal conductivity of perfect single crystal 6H-SiC and 3C-SiC predicted by first-principle calculations in Fig.3 (lines). The experimentally measured thermal conductivity of 6H-SiC from different research groups in the literature (Ref. 11, 15, 16 in the main text) match well with the theoretically

predicted thermal conductivity of perfect single crystal 6H-SiC (Ref. 17 in the main text). However, the experimentally measured thermal conductivity of 3C-SiC in the literature is significantly lower than the theoretically predicted thermal conductivity of perfect single crystal 3C-SiC. Therefore, we think the thermal conductivity reduction caused by extrinsic phonon scattering (purity and quality) in 6H-SiC is negligible while that for 3C-SiC is significant. This is what Mingo and coauthors pointed out in their theoretical paper (Ref. 1 in the main text). Additionally, the mature growth techniques and successful quality control of 6H-SiC crystals laid the foundation for current wide adoption of 6H-SiC electronics while the applications of 3C-SiC electronics are limited by the crystal purity and quality (Ref. 5 in the main text). This comparison further highlights the importance of our work which reports a high-purity and high-quality 3C-SiC crystals.

Moreover, we also perform first-principle calculations to calculate the intrinsic thermal conductivity of perfect single crystal 3C-SiC as a function of temperature, as shown in the figure below. The calculated thermal conductivity is much higher than the measured thermal conductivity in the literature and matches well with the measured thermal conductivity of our high-quality 3C-SiC bulk crystals (Fig. 3 in the main text).

Supplementary Fig. 13. First principle calculated thermal conductivity of perfect single crystal 3C-SiC.

In terms of the relation of B impurity and thermal conductivity, Ref. 1 in the main text studied all the measured thermal conductivity and impurity of 3C-SiC in the literature and found that the low thermal conductivity of 3C-SiC is due to phonon scatterings by impurity and grain boundary in the measured 3C-SiC crystals. Their theory showed the exceptionally strong resonant phonon scattering caused by B defects significantly reduced the thermal conductivity. Motivated by this theoretical work, we expect high thermal conductivity for high-purity and high-quality 3C-SiC crystals, especially with low concentrations of B impurity. We adopted the conclusion of B impurity and thermal conductivity from this theoretical paper (Ref. 1 the main text) to support our claim about the effect of B defects on thermal conductivity in this work.

To further prove that the B impurity reduces thermal conductivity significantly as predicted by the theoretical paper, we grow an additional sample (3C-SiC film on Si substrate) which is intentionally doped with B. The concentration of the intentional boron doping is $1-2 \times 10^{19}$ atoms cm^{-3} and the thickness of the 3C-SiC film is $1.87 \mu\text{m}$. The measured thermal conductivity of this doped sample is $324 \text{ W m}^{-1} \text{ K}^{-1}$ which is about 20% lower than the un-doped 3C-SiC film with a similar thickness ($1.75 \mu\text{m}$ in the main text). This reduction in thermal conductivity of 3C-SiC is consistent with the theoretical prediction in Ref. 1 in the main text, which further supports our claim about the B defects and thermal conductivity.

Revisions: We added the related discussions to the main text (Pages 3, 4, 12, and 13) and Supporting Information (SI) (Page 12).

Comment (2): Generally speaking, the thermal conductivity of thin film is usually higher than that of bulk sample due to size effect. The authors should assess the effect of small size and low dimension on the main conclusion of this work.

Response: We thank the reviewer for the comment. We interpret the reviewer's remarks to mean "lower" and not "higher" here since thin films usually have a lower thermal conductivity than bulk samples due to size effect instead of "higher". In this work, we studied both thin films and free-standing bulk 3C-SiC samples. The measured thermal conductivity of the free-standing bulk 3C-SiC sample agrees excellently with first-principle predicted values of perfect 3C-SiC single crystal.

We don't think there exists size effect in the free-standing bulk 3C-SiC crystals since the thickness of the free-standing bulk crystals (100 μm) is much larger than the dominant phonon mean free paths in 3C-SiC and thermal penetration depth in the TDTR measurements. For the 3C-SiC thin films grown on Si substrates, the measured thermal conductivity is lower than the measured thermal conductivity of the free-standing bulk sample and also shows strong thickness dependence due to size effect (Fig. 4) as expected. We reported the thermal conductivity of both free-standing bulk crystals and thin films so the small size and low dimension the reviewer mentioned does not affect the conclusions of this work.

To figure out the origin of the size effect in 3C-SiC thin films, we perform first-principle calculations to calculate the phonon dispersion relation of 3C-SiC and the phonon mean free path accumulated thermal conductivity in perfect 3C-SiC single crystal, as shown in the figure below. It shows the contributions of phonons with different mean free paths to the total thermal conductivity of perfect 3C-SiC single crystal. For phonons with mean free paths longer than the film thickness, the phonons scatter with the film boundaries which cause reduction in phonon mean free paths and corresponding thermal conductivity of thin films.

Supplementary Fig. 14. Calculated phonon properties of perfect 3C-SiC single crystal. a Phonon dispersion relation of 3C-SiC. **b** Accumulated thermal conductivity of perfect 3C-SiC single crystal at 300, 400, and 500 K. It shows the contributions of phonons with different mean free paths to the total thermal conductivity of perfect 3C-SiC single crystal.

Revisions: We added the related discussions to the main text (Pages 9, 15, and 17). We added the bulk value of 3C-SiC thermal conductivity into Fig. 4 as a comparison. We added the phonon

dispersion relation and phonon mean free path accumulated thermal conductivity into the SI (Pages 13 and 14).

Comment (3): The authors attributed high thermal conductivity of 3C-SiC to the lack of defect-phonon scatterings. However, there was characterization missing for the defects in the 3C-SiC samples.

Response: We thank the reviewer for the comment. Please note that we characterized the defect concentrations (boron, oxygen, and nitrogen impurities) of the 3C-SiC crystal from both two sides of the free-standing 3C-SiC wafer. The corresponding impurity concentrations are included in Page 6 in the main text. We found low concentrations of these impurities and their effects on thermal conductivity are negligible according to the theoretical calculations in Ref. 1 in the main text. The detailed SIMS data are also shown in Supplementary Fig. 3 in the SI.

Revisions: We added more discussions in the main text (Page 12) to highlight the measured low concentrations of impurities.

Comment (4): The labeling of the references in Figs. 2-5 is confusing and unclear to the readers.

Response: We thank the reviewer for the comment. We revised the labeling and we added a list of thermal conductivity of diamond thin films and the corresponding references to SI.

Revisions: We revised the labeling and added the list to the SI (Page 15).

Reviewer #2

In this manuscript, the authors report a variety of thermal conductivity measurements on high-purity bulk and thin-film 3C-SiC samples grown by low-temperature CVD. They find extremely high thermal conductivity values which are to be expected based on theory. However, as the authors point out, previous measurements on 3C-SiC materials (of presumably poorer quality) had

shown lower thermal conductivity than predicted by theory. Thus, this work appears to resolve that contradiction between the theoretical predictions and previous experiments.

The work is interesting (a) from a purely scientific standpoint (resolving the aforementioned contradiction between theory and experiments), and also (b) since 3C-SiC has the potential for widespread use in high power electronic devices due to its excellent thermal and electrical properties and compatibility with other important materials.

The work is complete with thermal conductivity measurements of bulk materials, thin-films of various thickness (both in-plane and cross-plane), and also as a function of temperature. Thermal boundary conductance measurements are also reported. The corresponding structural characterization is also extensive and complete. The thorough work makes for a convincing argument that these measurements are correct.

Overall, the manuscript is well-written with extensive references to the relevant literature. It should be well received and highly cited within the community. I only have a few small comments below which should be easy to address or dismiss.

Response: We thank the reviewer for all the positive evaluations of our manuscript as “interesting, complete, and well-written”.

Comment (5): Small point, but SAED should be defined when first used in line 109.

Response: We thank the reviewer for the comment. We agree that SAED should be defined first.

Revisions: We revised it according to the suggestion of the reviewer.

Comment (6): It might be worthwhile to state the detection limit in SIMS for boron in line 120. This would help readers place an upper limit on the boron concentration in the samples, which is helpful in context with the calculations from Ref. 1.

Response: We thank the reviewer for the comment. We agree with the reviewer.

Revisions: We added the detection limit in SIMS for boron (Page 7).

Comment (7): In lines 105, 184, 212, 327 “magnitude” should be “maximum”

Response: We thank the reviewer for the comment. We agree with the reviewer.

Revisions: We revised them accordingly.

Comment (8): This may well be outside the scope of the work, or proprietary company information, but as I read the manuscript I found myself thinking that other groups have likely (surely?) tried to create high quality (low boron) 3C-SiC before. So the natural question to me is, what was different in the growth of these materials that makes them so much better? A few sentences or a short paragraph regarding the improvements in growth would probably be welcome to most readers.

Response: We thank the reviewer for the comment. The samples in this work are grown in a customized CVD reactor in Air Water. Inc. after decades’ optimization and improvement in growth of high-quality crystals. It is not a standard operation since many known or unknown factors affect crystal quality. Some growth details are important according to the growers: the orientation of Si substrate and growth temperature. Since both Si and 3C-SiC have the same rotational symmetry (120°) about the [111] axis, (111) 3C-SiC layers can be grown on (111) Si substrates with low density of stacking faults and double positioning boundary at relatively low crystal growth temperature (1300 K). We noticed a recent paper which reported the measured thermal conductivity ($\sim 90 \text{ W m}^{-1} \text{ K}^{-1}$) of thermally thick 3C-SiC grown on (100) Si (purchased from MTI).¹ This reported thermal conductivity is significantly lower than that of our 3C-SiC samples.

We do not exactly know the growth details and CVD reactor structures of other groups so it is difficult to know the exact reason about the difference in crystal quality for now. But this does not

affect the reproduction of our manuscript. Anyone who would like to reproduce this manuscript can purchase samples from Air Water Inc.

Revisions: We added some discussions to the Methods section about the sample growth (Page 20).

Comment (9): The word “properties” (or a similar substitute) appears to be missing from line 279.

Response: We thank the reviewer for the careful checking. We agree with the reviewer.

Revisions: We made the correction in the manuscript.

Reviewer #3

This manuscript reported a high thermal conductivity in cubic-SiC. The value of over 500 W/mK at room temperature is probably the highest value for SiC till now. It also confirm the traditional theory that structural complexity and thermal conductivity are inversely correlated. I think that the paper can satisfy the journal after revision. Please find questions as follows:

Response: We thank the reviewer for the positive evaluation of our manuscript.

Comment (10): Frankly speaking, it is not certain that 3C-SiC is the only SiC polytype which can be grown on Si.

Response: We thank the reviewer for the comment. This information is from Ref. 5 in the main text. It is correct for now, to the best of our knowledge. But we agree with the reviewer that other polytypes of SiC may also be able to grow on Si in the future as the development of growth techniques in the future.

Revisions: We deleted the statement of “only” in the manuscript.

Comment (11): The 2 inch wafer of the 3C-SiC shows yellow color. What kind of defects or dopants make the wafer looks yellow? How does those defects influence the thermal conductivity?

Response: We thank the reviewer for the comment. The yellow color is from two reasons. First, the bandgap of 3C-SiC is 2.3 eV which corresponds to the energy of photons with wavelength of 539 nm. The intrinsic absorption of 3C-SiC makes it look yellow. Second, the nitrogen defects also possibly contribute to the yellow color. The concentration of nitrogen defect in the 2-inch 3C-SiC wafer is about 5.8×10^{15} atoms cm^{-3} . This low concentration of nitrogen defect has negligible effect on the thermal conductivity according to the calculations in Ref. 1 in the main text.

Revisions: We added related discussions in the main text (Page 6).

Comment (12): The author claimed that Boron greatly influenced the thermal conductivity of 3C-SiC. In my understanding, as a point defect, Boron defects usually don't degrade the crystal quality much, in particular the values of XRD omega-rocking. In other words, a high concentration of boron does not prevent SiC as a single crystal. Therefore it is better to make it clear that the excellent value reported in this work is due to the single crystallinity or the quite low concentration of Boron?

Response: We thank the reviewer for the comment. We agree with the reviewer that a high concentration of boron does not prevent SiC as a single crystal. But heterogeneous epitaxial growth of 3C-SiC in the literature can lead to grain boundaries and stacking faults which reduce thermal conductivity. These two factors (crystal quality and crystal purity) affect thermal conductivity simultaneously. The high thermal conductivity reported in this work is due to both the single crystallinity and the low concentrations of impurities. The detailed theoretical prediction of the relation between thermal conductivity and impurity and crystal quality can be found in Ref. 1 in the main text.

To further prove that the B impurity reduces thermal conductivity significantly as predicted by the theoretical paper, we grow an additional sample (3C-SiC film on Si substrate) which is intentionally doped with B. The concentration of the intentional boron doping is $1-2 \times 10^{19}$ atoms

cm⁻³ and the thickness of the 3C-SiC film is about 1.87 μm. The measured thermal conductivity of this doped sample is 324 W m⁻¹ K⁻¹ which is about 20% smaller than the un-doped 3C-SiC film with a similar thickness (1.75 μm in the main text). This reduction in thermal conductivity of 3C-SiC is consistent with the theoretical prediction in Ref. 1 in the main text.

Revisions: We added related discussions in the main text (Pages 12 and 13).

Comment (13): Nowadays, the single crystal diamond is larger than 1 inch.

Response: We thank the reviewer for the comment. We double checked literature. Transparent polycrystalline diamond can achieve large wafers. But single crystal diamond is still about 1.5 inch in size according to a recent review paper from the ultrawide bandgap semiconductor community (Advanced Electronic Materials, 4(1), 1600501, 2018). We also noticed some more recent news release from CVD diamond companies such as Kenzan Diamond to report a 2-inch diamond wafer but lack of peer-review publication. If the reviewer can provide reference or literature for larger size of single crystal diamond wafers, we are willing to update the data in the manuscript.

Revisions: We updated the single crystal diamond wafer size to 2 inches.

Comment (14): In page 13, ‘thin films such AlN’ should be ‘thin films such as AlN’

Response: We thank the reviewer for the comment. We agree with the reviewer.

Revisions: We revised it in the manuscript.

Comment (15): In fig.4, the authors compared the cross-plane thermal conductivity of different kinds of semiconductors with varied thicknesses. What is the reason for the thickness-dependent thermal conductivity? Why do the 3C-SiC, AlN, GaN and diamond show different tendency?

Response: We thank the reviewer for the comment. Thickness dependent thermal conductivity is due to size effect from the film boundaries which scatter phonons. Phonons with mean free paths

longer than the film thickness scatter with the film boundaries, which reduces the phonon mean free paths and corresponding thermal conductivity of films. The thinner the films are, the more extensive the phonons scatter with the boundaries. The different trends of thickness dependent thermal conductivity of different semiconductors are from the intrinsically different phonon mean free path distributions in these semiconductors. We perform first principle calculations to calculate the phonon mean free path distribution of perfect 3C-SiC single crystal and compare with the phonon mean free path distributions of other semiconductors. The accumulated thermal conductivity of perfect single crystal 3C-SiC, AlN, GaN, and diamond scaled by their bulk thermal conductivity at room temperature are shown in the figure below. The data of AlN, GaN, and diamond are from literature.²⁻⁴ Please note that hetero-epitaxial growth of diamond films are polycrystalline so the thermal conductivity of diamond films (Fig. 4 in the main text) are affected by grain boundary and other structural imperfections as well.

Supplementary Fig. 15. The accumulated thermal conductivity of 3C-SiC, AlN, GaN, and diamond single crystals scaled by their bulk thermal conductivity at room temperature.

Revisions: We added related discussions to the main text (Pages 9, 15, and 17) and SI (Pages 13 and 14).

Comment (16): In fig. 5, The author compared the thermal boundary conductance of different kinds of interface. It is known that the boundary conductance is greatly related the interface roughness? How about the interface sharpness of those interfaces?

Response: We thank the reviewer for the comment. We agree with the reviewer that thermal boundary conductance is related to interface roughness. All the interfaces included in Fig. 5 are smooth interfaces (sub-nm roughness). Only the GaN-BAs interface is a bonded interface with a thin amorphous interfacial layer while all the other interfaces are fabricated by growing one semiconductor on the top of the other well-polished semiconductor substrates.

Revisions: We added related discussions in the main text (Page 19).

References

- 1 Khan, S. *et al.* Properties for Thermally Conductive Interfaces with Wide Band Gap Materials. *ACS Appl. Mater. & Interf.* **14**, 36178-36188 (2022).
- 2 Qian, X., Zhou, J. & Chen, G. Phonon-engineered extreme thermal conductivity materials. *Nat. Mater.* **20**, 1188-1202 (2021).
- 3 Li, W. *et al.* Thermal conductivity of diamond nanowires from first principles. *Phys. Rev. B* **85**, 195436 (2012).
- 4 Koh, Y. R. *et al.* Bulk-like Intrinsic Phonon Thermal Conductivity of Micrometer-Thick AlN Films. *ACS Appl. Mater. & Interf.* **12**, 29443-29450 (2020).

REVIEWERS' COMMENTS

Reviewer #1 (Remarks to the Author):

To the Editors/Authors,

I am sorry to the late reply.

Thanks for the detailed replies from the authors. The response was adequate. I have a little question.

In the response of comment (3), the authors characterized the defect concentrations (boron, oxygen, and nitrogen impurities) of the 3C-SiC crystal. However, the intrinsic defects (i.e., point defects, dislocation, and interface, etc.), which might have negative effects on the thermal conduction of the crystal, were not present in the work.

Reviewer #2 (Remarks to the Author):

The authors have addressed all my comments and questions, and I believe that the paper is suitable for publication.

Reviewer #3 (Remarks to the Author):

The paper was well revised and satisfies the journal.

We sincerely thank the reviewers and the editor for the precious time and attention on our manuscript. We also greatly appreciate the constructive and valuable comments from the reviewers. These insightful suggestions and corresponding revisions improve our manuscript and make it more complete.

Reviewer #1:

Comment (1): Thanks for the detailed replies from the authors. The response was adequate. I have a little question. In the response of comment (3), the authors characterized the defect concentrations (boron, oxygen, and nitrogen impurities) of the 3C-SiC crystal. However, the intrinsic defects (i.e., points defects, dislocations, and interface, etc.), which might have negative effects on the thermal conduction of the crystal, were not present in the work.

Response: We thank the reviewer for the comment. We acknowledge that we only characterized the three main point defects (boron, oxygen, and nitrogen). The other point defects such as vacancies were not characterized because of technical difficulties. We expect low concentrations of all the other point defects. For extended defects, stacking faults are typically dominant in 3C-SiC compared with dislocations. The stacking fault density on the growth face of the bulk 3C-SiC is low (about 1000 cm^{-1}) which has negligible effect on thermal conductivity. We did not characterize the dislocation density. The bulk crystals are single crystal so there are no interfaces inside the crystals.

Revisions: We added the related discussions to the main text (Pages 6 and 7).